# Regulatory Efficacy of *Spirulina platensis* Protease Hydrolyzate on Lipid Metabolism and Gut Microbiota in High-Fat Diet-Fed Rats

**DOI:** 10.3390/ijms19124023

**Published:** 2018-12-13

**Authors:** Pengpeng Hua, Zhiying Yu, Yu Xiong, Bin Liu, Lina Zhao

**Affiliations:** 1College of Food Sciences, Fujian Agriculture and Forestry University, Fuzhou 350002, China; huapengpeng_flower@163.com; 2National Engineering Research Center of JUNCAO Technology, Fujian Agriculture and Forestry University, Fuzhou 350002, China; 18838018055@163.com (Z.Y.); aindxiong@163.com (Y.X.)

**Keywords:** *Spirulina platensis* protease hydrolyzate, peptides, lipid metabolism disorder, gene expression, gut microbiota

## Abstract

Lipid metabolism disorder (LMD) is a public health issue. *Spirulina platensis* is a widely used natural weight-reducing agent and *Spirulina platensis* is a kind of protein source. In the present study, we aimed to evaluate the effect of *Spirulina platensis* protease hydrolyzate (SPPH) on the lipid metabolism and gut microbiota in high-fat diet (HFD)-fed rats. Our study showed that SPPH decreased the levels of triglyceride (TG), total cholesterol (TC), low-density-lipoprotein cholesterol (LDL-c), alanine transaminase (ALT), and aspartate transaminase (AST), but increased the level of high-density-lipoprotein cholesterol (HDL-c) in serum and liver. Moreover, SPPH had a hypolipidemic effect as indicated by the down-regulation of sterol regulatory element-binding transcription factor-1c (SREBP-1c), acetyl CoA carboxylase (ACC), SREBP-1c, and peroxisome proliferator-activated receptor-γ (PPARγ) and the up-regulation of adenosine 5’-monophosphate (AMP)-activated protein kinase (AMPK) and peroxisome proliferator-activated receptorα (PPARα) at the mRNA level in liver. SPPH treatment enriched the abundance of beneficial bacteria. In conclusion, our study showed that SPPH might be produce glucose metabolic benefits in rats with diet-induced LMD. The mechanisms underlying the beneficial effects of SPPH on the metabolism remain to be further investigated. Collectively, the above-mentioned findings illustrate that Spirulina platensis peptides have the potential to ameliorate lipid metabolic disorders, and our data provides evidence that SPPH might be used as an adjuvant therapy and functional food in obese and diabetic individuals.

## 1. Introduction

The improvement in living standards worldwide and the increasing intake of poor quality food, at least from the nutritional point of view, have resulted in an increasing frequency of lipid metabolism disorder (LMD) [1]. LMD is a risk factor for obesity, hyperlipidemia, hyperglycemia, hypertension, fatty liver, cardiopathy, clinical syndrome, and other metabolic syndromes. It is also one of the most threatening public health problems in the world [2]. According to the statistics of the World Health Organization, the above-mentioned diseases are responsible for more than 18 million deaths annually. LMD is characterized by high levels of triglyceride (TG), total cholesterol (TC), and low-density-lipoprotein cholesterol (LDL-c), coupled with low levels of high-density-lipoprotein cholesterol (HDL-c) [3]. Dysregulation of lipid metabolism in the liver induces abnormal accumulation of lipids and the subsequent formation of lipid droplets, known as hepatosteatosis. Adipose tissue, an endocrine organ that secretes a number of adipokines known to mediate lipid metabolism, inflammation, and insulin sensitivity, is also critical in metabolic control [4]. Due to the medical importance of LMD, considerable research has been devoted to develop appropriate treatments. Although several drugs have been approved by the US Food and Drug Administration to treat obesity, their efficacy is often low and side effects are common [5,6]. Therefore, it is urgently necessary to develop a well-tolerated treatment with minimal side effects for obesity.

Recently, hydrolysate from several plants has been reported to have LMD-preventing effects. These plants include lucid ganoderma [7,8]. Several active anti-obesity ingredients present in plants have also been identified, such as Kefir peptides from lucid ganoderma and peptide 2 from Grifola frondosa. Many soy peptides have been identified to lower cholesterol and triglycerides, which can suppress fat synthesis and storage in different experimental systems [9]. These plant extracts and natural compounds have considerable potential to be further developed into effective therapies for LMD [10].

*Spirulina platensis*, belonging to the family Oscillatoriaceae [11], grows naturally in alkaline lakes [12], and it is a special formula consisting of active constituents, including minerals, vitamins and proteins, beta-carotene, tocopherols, and phenolic acids, exhibiting high anti-inflammatory and antioxidant activities [13,14], especially essential amino acids [15,16]. Therefore, it is used as a food supplement for human and feed additives for many animal species as well as birds and fishes. Moreover, *Spirulina platensis* and its highly active ingredient (C-phycocyanin) exhibits anti-inflammatory, immunomodulatory, hepatoprotective, nephroprotective, neuroprotective, antidiabetic, antigenotoxic, anti-hypertensive, and anticancer activities [17,18,19,20]. More recently, *Spirulina platensis* has received increasing attention due to their potential biological activities, including ACE-inhibitory, antioxidant, and LMD-preventing properties [21]. However, the effects of *Spirulina platensis* protease hydrolyzate (SPPH) on dysregulated lipid metabolism in liver and adipose tissue of diet-induced obesity have not yet been fully elucidated.

At the molecular level, the major transcription factors, such as PPARγ, SREBP-1c, and AMPK, have been implicated in the regulation of obesity. Previous studies have shown that the down-regulation of lipogenic proteins and up-regulation of lipolytic proteins mitigates obesity and dyslipidemia in a high calorie diet-induced rodent model [22,23]. PPAR family plays an essential role in lipid metabolism and is mainly expressed in adipose tissue, liver, and skeletal muscle, mediating obesity/anti-obesity signaling events. PPARα regulates the metabolism of lipids, carbohydrates, and amino acids, and it can be activated by ligands [24]. AMPK plays a significant role in lipogenesis and fatty acid oxidation through inactivation of ACC and carnitine palmitolytransferase-1 [25]. SREBP-1c is an important transcriptional factor involved in regulation of key enzymes of lipogenesis, including ACC and FAS. Targeting lipid metabolism has been considered as a potential and alternative strategy to combat obesity [26]. Several studies have shown that gut microbiota plays an important role in the effect of hyperlipidemia. The disturbance of compositions of gut microbiota (the ratio of Firmicutes to Bacteroidetes and endotoxin levels) can affect the gut barrier function and hepatic cholesterol metabolism by several pathways [27,28,29]. Microbial species are associated with changes in blood lipids. The abundance of particular bacterial genera is negatively correlated with the body mass index and TG, but positively correlated with the HDL-c level. It is widely accepted that drugs can make changes in the gut microbiota, leading to an impact on the body’s state.

In the present study, we investigated whether SPPH has an effect on diet-induced obesity. We also measured body weight, serum index, liver index, the expressions of genes involved in adipogenesis, and the compositions of gut microbiota.

## 2. Results

### 2.1. Characterization of Potent Major Compounds

The peptide sequences of SPPH were identified (Appendix A). Our study showed that 217 peptide sequences were found from SPPH. Peaks were observed at different retention times ranging from 1.96 min to 9.93 min [30]. Representative chromatograms and MS/MS spectra of high and low abundance peptides are shown (Appendix A).

### 2.2. Effect of SPPH on Body Weight and Serum Lipids of HFD-fed Rats

Table 1 shows the changes in the body weight of the rats during the 8-week experimental period. The average initial body weight was 223.52 ± 6.15 g, 224.28 ± 6.58 g, and 227.08 ± 9.84 g for NFD (normal fat diet), HFD (high-fat diet), and SPPH (*Spirulina platensis* protease hydrolyzate) groups, respectively. After a 4-week experimental period, the body weight of the HFD group was significantly increased compared with the NFD group (Table 1). Moreover, there was a statistically significant difference between the SPPH and HFD groups (*p* < 0.01). After 8 weeks, the average final body weight was 374.58 ± 16.20 g, 412.05 ± 19.21 g, and 354.09 ± 13.11 g for the NFD, HFD, and SPPH groups, respectively. The average body weight of the NFD, HFD, and SPPH groups was increased by 163.14 ± 14.82 g, 215.36 ± 23.70 g, and 169.83 ± 42.68 g, respectively. SPPH (150 mg/kg) significantly (*p* < 0.01) reduced the body weight after 8 weeks of oral administration compared with the HFD group, while there was no significant difference between the SPPH group and NFD group.

At week 0, there was no significant difference in blood lipid levels among the groups. After 4 weeks, the levels of serum TC, TG, and LDL-c were significantly decreased in the SPPH group compared with the HFD group (*p* < 0.01) (Figure 1A,B,D). The HDL-c level in the SPPH group was significantly increased compared with the HFD group (*p* < 0.01) (Figure 1C). After 8 weeks, similar findings were observed, showing that the levels of serum TC, TG, and LDL-c were significantly decreased in the SPPH compared with the HFD group (*p* < 0.01) (Figure 1A,B,D). However, the HDL-c level in the SPPH group was significantly higher compared with the HFD group after 8 weeks of SPPH treatment (*p* < 0.01) (Figure 1C). The results showed that the SPPH treatment at a dose of 150 mg/kg significantly affected the serum lipid profile.

### 2.3. Effect of SPPH on Liver Function and Hepatic Steatosis

To examine the effect of SPPH on biochemical changes, we determined the levels of TG, TC, HDL-c, LDL-c, AST, and ALT in the liver. HFD significantly increased the levels of hepatic TG, TC, LDL-c, AST, and ALT. After 4 weeks, the SPPH treatment group showed markedly reduced levels of TG, TC, LDL-c, AST, and ALT in the liver (Figure 2A,B,D–F). In addition, after 8 weeks, the hepatic HDL-c level in the SPPH group was decreased (*p* < 0.01) compared with the NFD group, while it was increased by approximately 50% compared with the HFD group (Figure 2C). These results could be attributed to the ability of SPPH to effectively suppress accumulation of hepatic TG, TC, and LDL-c in HFD-fed rats. Moreover, the levels of AST and ALT in the HFD group were significantly increased compared with the NFD group, indicating that the rat model of liver dysfunction was successfully established. However, SPPH treatment significantly reduced these two parameters (*p* < 0.01), indicating that SPPH could effectively improve liver function.

H&E staining showed the effect of SPPH on HFD-induced lipid accumulation in the liver (Figure 3). The liver color of the NFD group was red with a smooth surface, and the liver volume was moderate, accompanied with normal tissue elasticity. Rats in the HFD group had a white and pink liver, as well as a tense capsule with swelling tissue. Compared with the HFD group, the liver in the SPPH group had a rosy color in varying degrees and a greater tissue elasticity. The histopathological examination of the NFD group showed normal cell architecture, the liver cells had polygonal edges with clear cell borders, and the nucleus was round and clear, which was located in the centre of the cell with abundant cytoplasm. However, the HFD control group showed significant morphological changes with greater hepatic lipid accumulation and fatty degeneration, and some cell nuclei apparently exhibited a typical fatty degeneration. Most of the cells in the SPPH group were restored to the normal levels, liver cell cords were arranged normally, and the overall cell degeneration was significantly improved. The results showed that SPPH treatment had a better preventive effect.

### 2.4. Effect of SPPH on the Expressions of Genes Involved in Lipid and Glucose Metabolism at the mRNA Level

To reveal the molecular mechanism underlying the effects of SPPH treatment, we analyzed the expressions of genes involved in lipid homeostasis in the liver (Figure 4). We assessed hepatic levels of lipogenesis-related genes (SREBP-1c and PPARγ), as well as fatty acid metabolism-related genes (PPARα and ACC) in HFD-fed rats. Our data showed that the expression of SREBP-1c was significantly decreased in the SPPH group compared with the HFD group. Of these genes, SREBP-1c and PPARγ were the most significantly decreased ones in the SPPH group (*p* < 0.01) compared with the HFD group. The expressions of PPARα and ACC, the genes involved in fatty acid β-oxidation, were significantly decreased in the SPPH group compared with the HFD group (*p* < 0.01), which was similar to that observed in NFD-fed rats. Taken together, these results demonstrated that SPPH inhibited fatty acid synthesis and activated fatty acid β-oxidation in the livers of HFD-fed rats.

### 2.5. SPPH Modulates Gut Microbiota of HFD-Fed Rats

To investigate whether the gut microbiota profile was altered by SPPH treatment, we assessed the dominant microbiota population in the NFD, HFD, and SPPH groups (Figure 5). As expected, at the genus level, the alteration of gut microbiota composition by HFD feeding was significant (*p* < 0.0001). The gut microbiota population of the SPPH group was also significantly different from the HFD group (*p* < 0.05). In this study, *Lachnospiraceae*, *Bacteroides*, *Lactobacillus*, *Alistipes*, and *Ruminococcaceae* were dominant genera presented in different groups. Through 8 weeks of SPPH treatment, the relative abundances of *Porphyromonadaceae*, *Lachnospiraceae*, *Prevotella*, *Ruminococcaceae*, *Bacteroides*, *Blautia*, *Desulfovibrionaceae*, *Alloprevotella*, and *Porphyromonadaceae* were significantly changed, and *Ruminococcus* was the most prominently increased one after SPPH treatment at the genus level. Moreover, the abundances of *Allobaculum*, *Firmicutes*, *Clostridium_XlVa*, and *Lachnospiracea* were decreased in the SPPH group compared with the HFD group. In addition, the lower B/F ratio in the HFD group was increased by SPPH treatment. These results suggested that HFD feeding could dysregulate gut microbiota distribution, while SPPH treatment could partially restore the microbiota distribution to the level of the NFD group.

### 2.6. Correlations of Biochemical Data and Key Phylotypes of Caecal Microbiota

To explore the interactive features between the lipid metabolism-associated parameters and gut microbiota during the HFD-induced obesity development, correlation analysis was performed to examine the possible connection between the abundance of gut bacteria and host metabolic parameters (Figure 6). The microbes (such as *Turicibacter*, *Romboutsia*, *Phascolarctobacterium*, *Erysipelotrichaceae*, *Firmicutes*, and *Clostridium XVIII*) that were significantly enriched in the HFD group were positively correlated with the levels of serum TG, TC, and LDL-c, but negatively correlated with the serum HDL-c level.

The relative abundances of *Porphyromonadaceae* and *Desuifovibrionaceae* were positively correlated with the serum HDL-c level, while *Turicibacter* was negatively correlated with the serum HDL-c level. The relative abundances of *Coprococcus*, *Erysipelotrichaceae*, *Blautia*, *Allobaculum*, *Bifidobacterium*, *Romboutsia*, and *Phascolarctobacterium* were positively correlated with the serum LDL-C level, and the relative abundances of *Barnesiella*, *Oscillibacter*, and *Paraprevotella* were significantly negatively correlated with the serum LDL-c level. Meanwhile, the serum TC level was positively correlated with the relative abundances of *Olsenella*, *Bifidobacterium, Romboutsia*, and *Phascolarctobacterium*, but negatively correlated with *Barnesiella*, *Oscillibacter*, and *Paraprevotella*. Interestingly, body weight was positively correlated with *Olsenella*, *Allobaculum*, *Bifidobacterium*, and *Phascolarcto*, but negatively correlated with *Oscillibacter* and *Paraprevotella*. Moreover, the serum TG level was positively correlated with *Romboutsia* and *Phascolarcto*, but negatively correlated with *Barnesiella* and *Paraprevotella*, suggesting that these bacteria constituted an important factor in the beneficial effect of SPPH.

## 3. Discussion

As important components of metabolic syndrome, obesity and hyperlipidemia are present in the majority of patients with cardiovascular and cerebrovascular diseases, resulting in high mortality rates [31]. The prevalence of obesity in recent decades can be largely attributed to changes in food habits and increased sedentary life style. Despite the rapidly growing recognition of hepatic steatosis, therapy or prevention of the disease remains less available [32]. In view of the very limited availability of FDA approved anti-obesity drugs and considering their side effects, it is quite necessary to find novel and effective natural product-based drugs to combat obesity.

Among different targets to treat obesity, the ones that can interfere with the process of lipid mobilization are fundamental [33,34,35]. Previous studies have shown that TC, TG, HDL-c, and LDL-c are strongly correlated with the prevalence and incidence of metabolic syndrome and cardiovascular diseases [36,37]. The serum LDL-C and TG levels are considered key risk indicators for atherosclerotic cardiovascular disease [38]. An evidence-based study has also shown that lowering the serum LDL-C and TG levels can effectively ameliorate the risk of vascular disease and reduce the incidence of acute coronary events [39]. AST and ALT are investigated as markers of liver damage, regarding the changes in lipid of HFD-fed rodents [40]. Measurement of liver damage caused by fat accumulation is important for diagnosis of non-alcoholic fatty liver disease (NAFLD). Obesity causes altered function of adipocytes, leading to expanded adipocyte mass and increased release of FFAs in the blood. Excess accumulation of TG in the liver results in significant and more abundant lipid accumulation [41,42]. In our experiment, we established an animal model of HFD-induced obesity, which is considered to be a good model as it has been reported to bear close resemblance to human obesity [43]. Consistent with a previous study, we showed that HFD resulted in significantly increased body weight, elevated levels of TC, TG, LDL-C, AST, and ALT, and decreased HDL-c level in serum and liver. Moreover, there were fat accumulation and massive accumulation of lipid droplets in the liver. However, the weight-losing effect of SPPH treatment was significant on HFD-fed rats, which might be associated with enhanced energy metabolism [44]. These results provided new insights into the anti-obesity effect of SPPH. In addition, SPPH treatment resulted in decreased levels of TC, TG, HDL-c, and LDL-c in serum and liver compared with the HFD group, which were similar to the levels of the ND group. Meanwhile, SPPH treatment also resulted in a significant decrease in hepatocyte steatosis, lipid droplets, and hepatic lipid accumulation. We showed a significantly increased number and size of fatty hepatocytes upon HFD administration, while such changes returned to the normal levels in the SPPH group [45]. SPPH have anti-obesity and hepatoprotective capacities [46]. Previous studies have shown that body weight is reduced by extracts of a few plant species, including leafs of Murraya koenigii and resin of Commiphora mukul, in HFD-fed rats [47,48,49,50]. These results demonstrated SPPH could be used as an effective agent in ameliorating the HFD-induced effects.

We investigated the expressions of several genes related to fatty acid transport (PPARα and ACC) and lipid metabolism, including lipogenesis (SREBP-1c and PPARγ) and β-oxidation (AMPK), to explore the possible mechanism of SPPH in decreasing accumulation of liver lipids. Several studies have demonstrated that SREBP-1c, a major transcription factor involved in hepatic lipogenesis, plays an important role, leading to increased fatty acid synthesis as a result of the induction of ACC [51,52]. One study has reported that the expression of SREBP-1c is positively correlated with the degree of hepatic steatosis in NAFLD patients [53]. Results of the current study showed that the expression of SREBP-1c was significantly lower in the liver of the NFD group compared with the HFD-fed rats. Consistent with a previous study, SPPH treatment effectively inhibited such increased expression of SREBP-1c. The expression of SREBP-1c transcriptional target ACC was strongly correlated with the SREBP-1c expression, suggesting that suppression of ACC contributed to a reduction in lipid accumulation in fatty liver [51,52]. SPPH treatment decreased the HFD-induced high expression of ACC. Taken together, these results demonstrated that SPPH down-regulated the expressions of lipogenesis-related genes in HFD-induced fatty liver. Of particular importance, PPARα is a ligand-activated transcription factor, and its activation induces the expressions of several genes involved in fatty acid oxidation at the mRNA level to reduce the circulating lipid levels [54]. Meanwhile, the activation of intestinal fatty acid oxidation by PPARα agonist bezafibrate suppresses postprandial lipidaemia in rats [46], and PPARα reduces the plasma TG and TC levels. Our current results showed that PPARα expression was significantly lower in the HFD group compared with the NFD group, while it was increased by SPPH treatment. A recent report has indicated that PPARα also modulates the expressions of lipogenic genes in liver, such as ACC, which are closely related to fatty acid synthesis and oxidation in hepatic steatosis in HFD-fed animals [55]. Liver adipose tissues are associated with the pathogenesis of metabolic syndrome [56]. Collectively, our results showed a significantly smaller liver lipid droplet area in the SPPH group, accompanied by an increased expression of PPARα and decreased expressions of SREBP-1c and ACC. Fat cell formation or adipogenesis is a differentiation process, by which undifferentiated preadipocytes are converted in to fully differentiated adipocytes [57]. Adipose tissue is a dynamic organ, the mass of which changes during a lifetime in response to metabolic requirements of the animal, thus playing an important role in energy balance. Particularly, SREBP-1c, one of the pro-adipogenic transcription factors, induces the PPARγ expression and regulates the expressions of AMPK [58,59,60]. However, we clearly showed that administration of SPPH considerably affected SREBPs. SREBPs are another family of transcription factors, but they are majorly involved in the regulation of lipid homeostasis by activating the expressions of genes required for the synthesis and uptake of cholesterol, fatty acid, and triglycerides. Previous studies have suggested that AMPK plays a role in the physiological regulation of fatty acid and glucose metabolism as well as in the regulation of appetite [61]. In our study, we found that AMPK was down-regulated in the HFD group. However, such alterations in the HFD-fed group were considerably reversed by the SPPH treatment. These results suggested that SPPH could decrease body weight and fat mass through down-regulation of SREBP-1c and PPARγ, which in turn resulted in inhibited expressions of lipogenic enzymes. Furthermore, we proved the anti-obesity activity of SPPH as evidenced from the reduced size of adipocytes in the SPPH group. Previous studies have demonstrated the anti-diabetic and anti-hyperlipidemic activities of elllagic acid, betulinic acid, and arjunolic acid from different plant sources [62,63].

Gut microbiota can directly affect blood cholesterol levels with its effects on the development of atherosclerosis, and several studies have proved the effects of an HFD on gut microbiota [64,65]. We compared the faecal and caecal microbiota of rats in three different groups to elucidate the precise underlying mechanism of improved hyperlipidaemia by SPPH. SPPH treatment increased the abundances of *Peptococcaceae*, *Prevotella*, *Alistipes*, *Porphyromonadaceae*, *Barnesiella*, and *Parasutterella*. *Prevotella* and *Alistipes* showed a negative correlation with the levels of serum TG, TC, and LDL-c, while they were positively correlated with the serum HDL-c level. The enterotype-like clusters driven by *Alistipes* and *Prevotella* (P-type) microbiota are characterized by a more conserved bacterial community [66]. The latest study has shown that there is a positive correlation between bile acid and *Prevotella. Prevotella* regulates lipid levels by altering the bile acid metabolism to change the blood lipid levels [67]. Our findings were consistent with a previous study. The abundances of *Alistipes* and *Prevotella* were increased by SPPH treatment. On the other hand, we know that the genus, *Barnesiella*, a family of *Porphyromonadaceae*, is part of the gut microbiota. In addition to the family, *porphyrinaceae*, the *Bacteroidetes* include the families, *Bacteroidaceae* and *Prevotellaceae*. Moreover, *Barnesiella* spp. regulates the composition of microbiota and optimizes host survival [68]. Besides, *Alloprevotella* and *Ruminococcus* were also enriched by SPPH treatment. These bacteria are short-chain fatty acid (SCFA) producers and negatively correlated with NAFLD and LMD [69]. Anaerobic bacteria are colonized in the cecum and colon, and they can ferment the non-digestible carbohydrate into SCFAs, such as propionate and butyrate. The SCFAs can be directly absorbed by the intestine and regulate the energy metabolism and insulin sensitivity of peripheral tissues via G protein-coupled receptors [70]. In addition, our findings showed that SPPH decreased the proportion of *Firmicutes* and increased the proportion of *Bacteroidetes* in caecal contents. These results were in accordance with the theory that the proportion of body fat is positively correlated with the abundance of *Firmicutes* in the gut microbiota in humans and mice. Besides, changes in body weight and serum LDL-c levels were positively correlated with *Firmicutes*. There were increased proportions of *Porphyromonadaceae*, which has been previously associated with NAFLD, atherosclerosis, and diabetes [71]. In addition to the increase in health-promoting bacteria, a loss of HFD-enriched microbes (Clostridium XVIII) was also observed in the SPPH-triggered alleviation of hyperlipidaemia. *Turicibacter*, belonging to the phylum, *Firmicutes*, may have a negative effect on gut health and the metabolic parameters in serum, such as TC and TG. Previous studies have shown that the abundance of *Clostridium XVIII* is increased in individuals with gastrointestinal disorders and dysfunctions, and such increased abundance of *Clostridium* may be induced by obesity-related metabolic disorders or pro-inflammatory responses [72]. However, the relationship between *Clostridium XVIII* and lipid metabolism remains largely unexplored. Therefore, our data directly proved that the HFD could cause colonic pathology and inflammation in HFD-fed rats, which might be associated with a proportional increase in *Clostridium XVIII*. In summary, we provided convincing evidence for the potential use of SPPH in hyperlipidemia and demonstrated that potent modulation of the intestinal microbiota during attenuation of metabolic disease was associated with its beneficial effects. Appendix A illustrates the mechanism by which SPPH reduced blood lipid levels. SPPH had the potential to ameliorate LMD, in part through modulating specific gut microbiota and regulating the expressions of the genes involved in lipid and cholesterol metabolism at the mRNA level. Therefore, SPPH might be beneficial for anti-hyperlipidemia, leading to the reduced risk of LMD.

## 4. Materials and Methods

### 4.1. Chemicals and Materials

In the present study, the air-dried and clean Spirulina platensis powder with a protein content of 60% was obtained from King Dnarmsa Spirulina Co., Ltd. (Fuqing, China). Protamex was purchased from Suo Laibao Biotechnology Co., Ltd. (Beijing, China).

### 4.2. Preparation of SPPH

*Spirulina platensis* powder and 95% ethanol (1:10, *W_powder_:V_ethanol_*) were mixed at 45 °C for 0.5 h, the macerate was filtered through Whatman filter paper No 3, then the supernatant was discarded in order to remove soluble substances in organic reagents, and the solid material was dried [73]. The dried solid material was soaked in 55% ethanol at a ratio of 1:10 (*w*/*v*) at 45 °C for 0.5 h, the macerate was filtered through Whatman filter paper No 3, the supernatant was then discarded in order to remove soluble substances in organic reagents, and the solid material was dried. The dried solid material was soaked in distilled water at a ratio of 1:10 (*w*/*v*) at 45 °C for 0.5 h, the macerate was filtered through Whatman filter paper No 3, the supernatant was then discarded in order to remove soluble substances in water, and the solid material was dried. The dried solid material was soaked in distilled water at a ratio of 1:10 (*w*/*v*), and the pH was adjusted to 7.5 prior to the addition of protamex (E:S = 1:50) (Enzyme: Substrate=1:50 (*m/m*)) and maintained at pH 7.5 and 45 °C. After 1 h, the hydrolyzed solution was bathed in boiling water for 10 min to inactivate the enzyme [74]. The supernatant was passed through a 100-µm mesh in order to remove solids in suspension. Subsequently, the supernatant was concentrated at 6000 g and 4 °C for 30 min in order to remove undigested proteins and inactivate the enzyme. Finally, the supernatant of SPPH containing the target *Spirulina platensis* peptides was collected and stored at −20 °C prior to further analysis. The obtained dry substance was named as SPPH. Meanwhile, *Spirulina platensis* peptides were prepared under the conditions as follows. The peptide in the sample was dissolved, followed by a desalting procedure. First, 300 µL of 50 mM ammonium bicarbonate was added to SPPH, the mixture was shaken and centrifuged at 12,000 g for 10 min, and the supernatant was transferred to a new EP tube. Then, 300 µL of 50 mM ammonium bicarbonate was added to the sample, the peptide was reconstituted once, and the supernatants were combined for two times. Second, 1 mL of 50% acetonitrile was added to the Bond Elut C18 desalting column and allowed to flow slowly through the desalting column. The desalting column was washed four times by adding 2% acetonitrile (0.1% formic acid) to a desalting column. The sample was made up to 1 mL with 2% acetonitrile (0.1% formic acid) and loaded onto a desalting column. The desalting column was washed four times with 2% acetonitrile (0.1% formic acid). The sample was eluted by adding 700 µL of 60% acetonitrile (0.1% formic acid) to the desalting column, and the eluate was collected. The sample after centrifugation and concentration was re-dissolved in the RPLC mobile phase A (0.1% formic acid, 2% acetonitrile/water) and bottled for online HPLC-MS/MS analysis [74].

### 4.3. HPLC-MS/MS Analysis of SPPH

The liquid phase was an ultra-fast liquid chromatograph Nexera XR (Shimadzu Corporation, Japan). The analytical column was a Waters BEH C18 column (1.7 µm, 2.1 × 50 mm) (Macherey-Nagel, Düren, Germany). The mobile phase consisted of binary mixture of 0.1% (*v*/*v*) formic acid (solvent A) and acetonitrile (solvent B). The flow rate was set at 0.3 mL/min. Elution gradient was as follows: 0–1 min: 5% B, 1–7 min: 5–60% B, 7–8 min: 60–80% B, 8–10 min: 5% B. The column temperature was set at 35 °C. The injection volume was 10 µL. TripleTOF 5600 system (AB SCIEX) equipped with a positive electrospray ionization source (ESI) was used. The MS parameters were set as follows: Spray voltage of 5600 V, air curtain pressure of 35 PSI, atomization pressure of 55 PSI, auxiliary gas of 50 PSI, ion source temperature of 500 °C, and solvent voltage of 100 V. The mass spectrometry scanning mode was the information-dependent acquisition mode (IDA Information Dependent Analysis). The first-level TOF-MS scanning ranged from 300–1500 *m*/*z*, the cumulative time was 250 ms, and the maximum charge (2+ to 5+) was 35 for each IDA cycle and a secondary map with a single-second count greater than 160 cps. The cumulative time for each secondary map was 60 ms, and the secondary mass spectrum scanning ranged from 100–500 *m*/*z*. Each cycle time was fixed at 2.5 s. The collision chamber energy setting was applied to all precursor ion collision-induced dissociation (CID), and the collision energy was automatically optimized. The dynamic exclusion was set to 6 s. The original wiff map file collected by mass spectrometry was processed and searched by PEAKS Studio 8 software (Bioinformatic Solutions Inc., Waterloo, ON, Canada). The databases were *Spirulina platensis* protein databases under Uniprot. The search parameters were set as follows: Cysteine alkylation to iodoacetamide modification, trypsin digestion, primary mass spectrometry mass tolerance was 20 ppm, secondary mass spectrometry was 0.1 Da, and peptide score (−10 lgP) greater than or equal to 20 was considered reliable.

### 4.4. Animals and Experimental Design

Thirsty healthy male Wistar rats aged 4 weeks were taken from Shandong Laboratory Animal Center of Shandong Academy (Shandong, China). All animal experiments procedures received care according to institutional guidelines, and all of the experiments were approved by the Fuzhou General Hospital Institutional Animal Care and Use committee. (IACUC approval no. CGU11-119). The animals were housed in a temperature-controlled room at 25% and 60% relative humidity with ad libitum access to food and distilled water and maintained on a reverse 12 h light/dark cycle during the experimental period. After 1 week of the acclimation period, rats were randomly divided into the following three groups stochasticly, normal fat diet (NFD) group (rats fed NFD, *n* = 8), high-fat diet (HFD) group (rats fed HFD, *n* = 8), and SPPH group (HFD-fed rats treated with SPPH, *n* = 8), by using a method described previously [75] Rats in the NFD group were given a basal diet (13.5% energy from fat; Lab Diet 5001; Laboratory Rodent Diet), and rats in the HFD and SPPH groups were given HFD (67% normal diet, 20% sugar, 10% lard, and 3% cholesterol). NFD and HFD groups were fed basal diet and HFD with 2 mL 0.9% saline orally, respectively, while the SPPH group was fed HFD with 2 mL SPPH extract (150 mg/kg·day) orally through gavage at the same time in the morning. The compositions of the experimental diets were based on the AIN-93 semisynthetic diet (American Institute of Nutrition, 1993, 1994).

### 4.5. Serum Samples Preparation

After 8 weeks of the experiment period, the rats were sacrificed following a 12 h fast and were anesthetized by intraperitoneal injection using ketamine hydrochloride, blood was collected from the heart, and the sample was transferred to a centrifuge tube. Serum was separated at 12,000 g for 10 min at 4 °C and was then stored at −80 °C until analysed.

### 4.6. Liver Homogenate Preparation

Liver was dissected, weighed, and cut into several sections, washed in saline solution (0.1 g of liver tissue was mixed with 0.9 mL of saline), dried, and immediately frozen and stored at −80 °C livers, after snap-freezing in liquid nitrogen. After centrifugation at 8000 *g* for 15 min at 4 °C, the supernatant was taken for analysis.

### 4.7. Biochemical Assays of Serum and Liver Tissue

The level of TC, TG, HDL-c, and LDL-c in rats’ serum using the assay kits (Nanjing Jiancheng Institute of Biotechnology, Nanjing, China) [76]. The level of TC, TG, HDL-c, LDL-c, ALT, and AST were measured in liver tissue using the corresponding assay kit. (Nanjing Jiancheng Institute of Biotechnology, Nanjing, China).

### 4.8. Liver Histopathological Analysis

The liver tissues were removed from each mice and samples were subsequently fixed in 4% 128 (*v*/*v*) paraformal dehyde/PBS then treated with ethanol solution. After that, all liver samples were fixed in paraformal dehyde and embedded in paraffin for staining with hematoxylin and 129 eosin or Oil red O, respectively. The slices were sectioned at 5 μm. The sections were observed for morphological evaluation at high magnification under an optical microscope (Nikon Eclipse TE2000-U, Nikon, Tokyo, Japan) [77,78].

### 4.9. mRNA Preparation and Gene Expression

Total tissue and cellular RNA was prepared by using Sepasol Super-I (Nacalai Tesque) according to the manufacturer’s instructions. cDNA was synthesized using PrimeScript™ RT reagent Kit with gDNA Eraser (Takara, Japan). RT-qPCR of AMPK-α, SREBP-1c, HMG-CoA, PEPCK, ACC, and control β-actin use the SYBR^®^ Premix Ex Taq™ II (Takara, Japan) to Monitor Gene Expression Levels. Following is a list of specific primers: AMPK-α, 5′-ATTTGCCCAGTTACCTCTTTCC-3′, R: 5′-GCTTGGTTCATTATTCTCCGAT-3′; SREBP-1c, F: 5′-GCTGTTGGCATCCTGCTATC-3′, R: 5′-TAGCTGGAAGTGACGGTGGT-3′); HMG-CR, F: 5′-AGTGGTGCGTCTTCCTCG-3′, R: 5′-CGAATCTGCTGGTGCTAT-3′); PEPCK, F: 5′-GAAAGTTGAATGTGTGGGTGAT-3′, R: 5′-TTCTGGGTTGATGGCCCTTA-3′; ACC, F: 5′-ACACTGGCTGGCTGGACAG-3′, R: 5′-CACACAACTCCCAACATGGTG-3′, and control β-actin, F: 5′-ACGTCGACATCCGCAAAGACCTC-3′, R: 5′-TGATCTCCTTCTGCATCCGGTCA-3′. Amplifications were performed using the AB7300 Real-Time PCR system (Waltham, USA). The conditions were as follows: Initial activation at 95 °C for 30 s, followed by 40 cycles of denaturation at 95 °C for 5 s, annealing at 60 °C for 31 s, and extension at 72 °C for 30 s. All data indicating mRNA expression levels are presented as a ratio relative to a control in each experiment Using an RNA extraction kit (Takara, Tokyo, Japan) to extract total RNA from the liver tissues, determination of the relative levels of target mRNAs was performed using the 2^−∆∆*C*t^ method and normalization [79].

### 4.10. Dynamic Profile of Intestinal Microflora in Response to SPPH

Caecal contents samples were collected at 8 weeks from rats from different groups, using a QIAamp-DNA stool mini kit (Qiagen, Hilden, Germany) to extract Metagenomic DNA from caecal contents of rats. The V3-V4 hypervariable regions of 16S rRNA gene from caecal microbiota were amplified using specific primers (F: 5′-CCTACGGRRBGCASCAGKVRVGAAT-3′ and R: 5′-GGACTACNVGGGTWTCTAATCC-3′) [80]. Sequencing was performed using a 2 × 300 paired-end (PE) configuration. Analysis was performed by MiSeq control software. The initial classification analysis was conducted on an Illumina’s Base Space cloud computing platform.

## 5. Bioinformatics Analysis

High-quality sequences were assigned to samples based on barcodes. In order to study the diversity information of species composition, the valid sequences were denoised. Results were generated by using Usearch (Version 7.1, http://drive5.com/uparse/) with 3% disagreement [81].

## 6. Statistical Analysis

The data for each group was expressed as mean ± standard deviation (SD). Statistical significance was measured using one-way analysis of variance (ANOVA). Statistical significance is expressed by a *p*-value less than 0.05. Relationships between gut microbiota composition and biochemical indicators in serum were determined using the Spearman’s rank correlation method.

## 7. Conclusions

Taken together, SPPH was able to affect the lipid metabolism of HFD-fed Wistar rats. After 8 weeks of SPPH treatment, body weight was increased, the levels of TC, TG, LDL-c, AST, and ALT in serum and liver were elevated, the liver steatosis was reduced, and the level of HDL-c was induced. Moreover, SPPH reduced the incidence of liver lesions and improved hepatocyte abnormality. SPPH supplementation directly affected lipid metabolism in the liver by influencing relational mRNA expression and affecting the gut microbiome by HFD-induced lipid metabolism disorder in rats. Collectively, we, for the first time, characterized a new potential therapeutic role of SPPH. However, we should conduct an in-depth analysis of changes in gut microbiota, and its underlying molecular mechanism upon SPPH supplementation should also be further assessed.

## Figures and Tables

**Figure 1 ijms-19-04023-f001:**
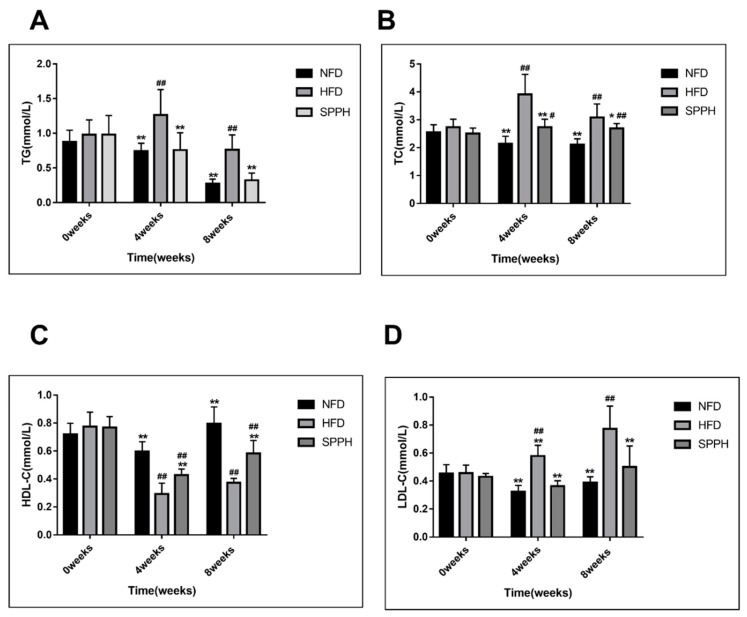
Serum lipid levels of rats in each group during the experimental period. TG (**A**), TC (**B**), HDL-c (**C**), LDL-c (**D**). Note: NFD, normal fat diet group; HFD, high-fat diet group; SPPH, HFD-fed rats treated with SPPH; NFD group, rats fed with NFD and gavaged with 150 mg/(kg·day) normal saline. HFD group, rats fed with HFD and gavaged with 150 mg/(kg·day) normal saline. SPPH group, rats fed with HFD and gavaged with 150 mg/(kg·day) *Spirulina platensis* protease hydrolyzate. ^#^
*p* < 0.05, ^##^
*p* < 0.01 vs. the NFD group; * *p* < 0.05, ** *p* < 0.01 vs. the HFD group.

**Figure 2 ijms-19-04023-f002:**
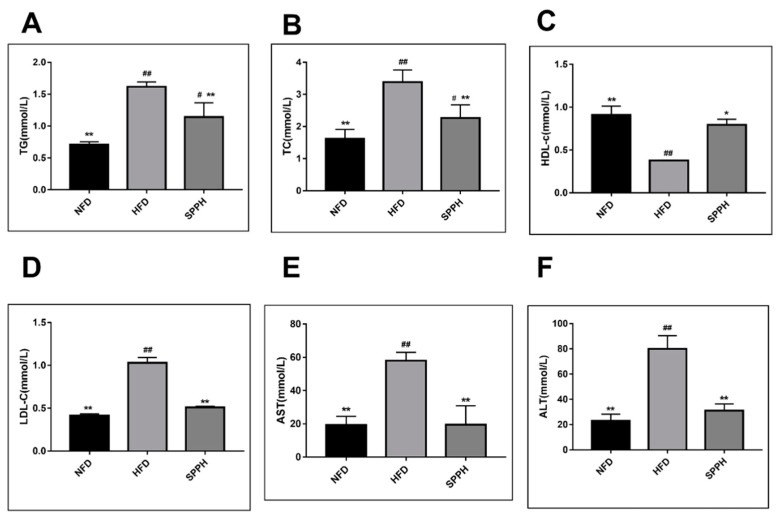
Hepatic lipid levels of rats in each group during the experimental period. TG (**A**), TC (**B**), HDL-c (**C**), LDL-c (**D**), AST (**E**), ALT (**F**). Note: TG, triglyceride; TC, total cholesterol; HDL-c, high-density-lipoprotein cholesterol; LDL-c, low-density-lipoprotein cholesterol. Data are expressed as mean ± SD (n = 8). Data are expressed as mean ± SD (n = 8). ^#^
*p* < 0.05, ^##^
*p* < 0.01 vs. the NFD group; * *p* < 0.05, ** *p* < 0.01 vs. the HFD group.

**Figure 3 ijms-19-04023-f003:**
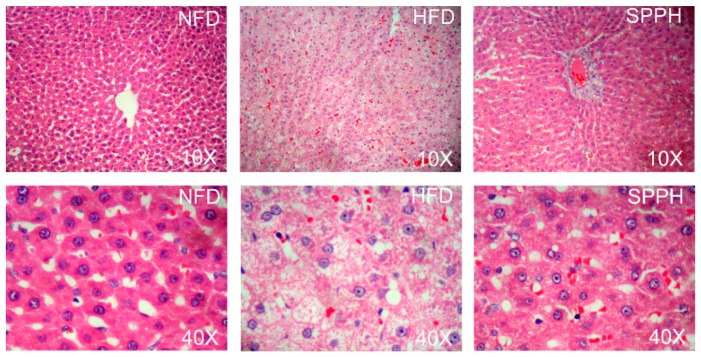
Histopathological analysis of rat hepatic tissues in different groups at 40× magnification.

**Figure 4 ijms-19-04023-f004:**
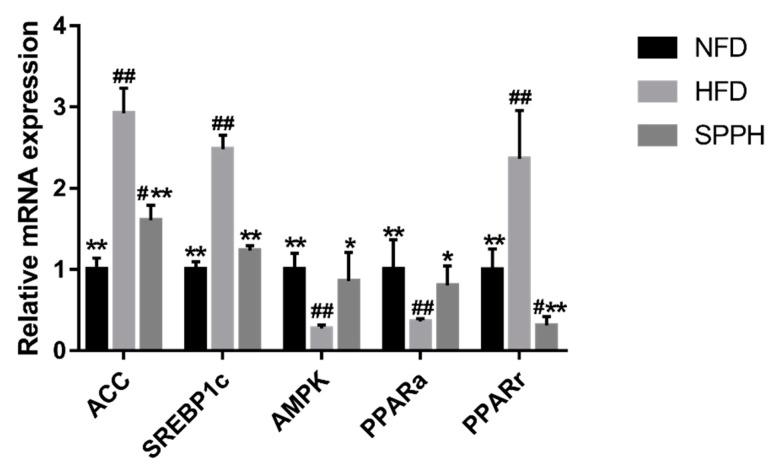
The mRNA and protein expressions levels of genes involved in lipid and glucose metabolism as determined using real-time PCR. Note: The differences were assessed by ANOVA and denoted as follows: ^#^
*p* < 0.05, ^##^
*p* < 0.01 vs. the NFD group; * *p* < 0.05, ** *p* < 0.01 vs. the HFD group.

**Figure 5 ijms-19-04023-f005:**
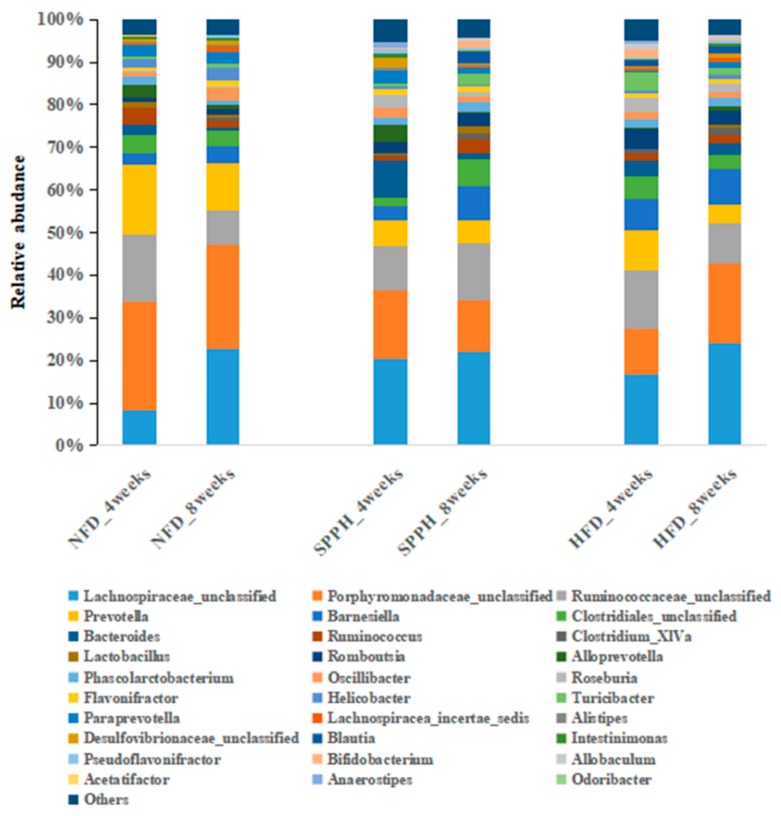
Changes in the bacterial composition of rat intestinal contents according to different genera. Composition of gut microbiota at the genus level.

**Figure 6 ijms-19-04023-f006:**
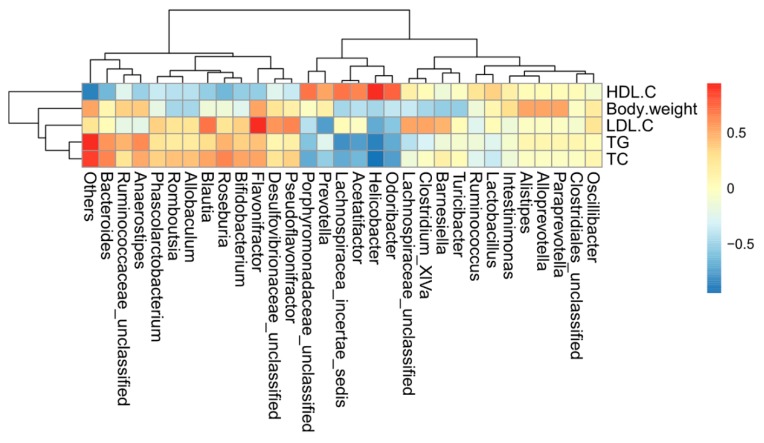
Statistical Spearman’s correlations between the caecal microbiota of significant differences and lipid metabolic parameters in SPPH, HFD, and NFD groups. The intensity of the color represents the degree of association between caecal microbiota of significant differences and MetS-associated parameters.

**Table 1 ijms-19-04023-t001:** Changes in the body weight of rats in the different groups during the experimental period.

Time	Weight (g)		
	NFD	HFD	SPPH
0 Weeks	223.52 ± 6.15	224.28 ± 6.58	227.08 ± 9.84
4 Weeks	382.69 ± 31.65 *	418.53 ± 18.48 ^#^	363.02 ± 42.65 **
8 Weeks	374.58 ± 16.20 **	412.05 ± 19.21 ^##^	354.09 ± 13.11 **
Weight gain	163.14 ± 14.82 **	215.36 ± 23.70 ^##^	169.83 ± 42.68 **

Note: NFD, normal fat diet; HFD, high-fat diet; SPPH, *Spirulina platensis* protease hydrolyzate. NFD group, rats fed with NFD and gavaged with 150 mg/(kg·day) normal saline. HFD group, rats fed with HFD and gavaged with 150 mg/(kg·day) normal saline. SPPH group, rats fed with HFD and gavaged with 150 mg/(kg·day) *Spirulina platensis* protease hydrolyzate. Data are expressed as mean ± SD (*n* = 8). ^#^
*p* < 0.05, ^##^
*p* < 0.01 vs. the NFD group; * *p* < 0.05, ** *p* < 0.01 vs. the HFD group.

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
