# Peer review of "Regulatory Efficacy of Spirulina platensis Protease Hydrolyzate on Lipid Metabolism and Gut Microbiota in High-Fat Diet-Fed Rats"

_ijms, 2018, doi:10.3390/ijms19124023_

Round 1
Reviewer 1 Report
This article reports effects of lipid extracts of Spirulina on lipid metabolism in high-fat diet rats. Splirulina extracts improved dyslipidemia in HFD fed rats by down regulating SREBP-1c, and upregulating AMPK-α in liver. In addition Splirulina extracts altered gut microbiota.
Although the study is well conducted, and results clearly presented and interpreted, there is no me mechanistic evidence. Authors should address this
Author Response
Response to reviewer 1
1. high-fat diet rats. Splirulina extracts improved dyslipidemia in HFD fed rats by down regulating SREBP-1c, and upregulating AMPK-α in liver. In addition Splirulina extracts altered gut microbiota. Although the study is well conducted, and results clearly presented and interpreted, there is no me mechanistic evidence. Authors should address this.
Response: Thank you very much for your positive comments.
Yes, SREBP-1c is a major transcription factor involved in hepatic lipogenesis, which leads to increases in fatty acid synthesis as a result of the induction of ACC. AMPK plays a significant role in lipogenesis and fatty acid oxidation through inactivation of ACC and carnitine palmitolytransferase-1. Several studys have reported that the level of SREBP-1c has positive correlation with the degree of hepatic steatosis in NAFLD patients. Results of the current study showed that the level of SREBP-1c was significantly higher in the liver of rats fed an HFD compared with that of ND rats. Also, cited references (Zou, B.; Ge, Z. Z.; Zhang, Y., et al. Persimmon tannin accounts for hypolipidemic effects of persimmon through activating of AMPK and suppressing NF-κB activation and inflammatory responses in high-fat diet rats [J]. Food & Function, 2014, 5(7):1536-1546. Yang, M.; Li, X.; Zeng, X., et al. Rheum palmatum L. Attenuates High Fat Diet-Induced Hepatosteatosis by Activating AMP-Activated Protein Kinase [J]. The American Journal of Chinese Medicine, 2016, 44(03):14. Song, Y.; Lee, S. J.; Jang, S. H., et al. Sasa borealis Stem Extract Attenuates Hepatic Steatosis in High-Fat Diet-induced Obese Rats [J]. Nutrients, 2014, 6(6):2179-2195. Tzeng, T. F.; Lu, H. J.; Liou, S. S., et al. Cassia tora (Leguminosae) seed extract alleviates high-fat diet-induced nonalcoholic fatty liver.[J]. Food & Chemical Toxicology, 2013, 51(1):194-201. Guo, W. L.; Pan, Y. Y.; Li, L., et al. Ethanol extract of Ganoderma lucidum ameliorates lipid metabolic disorders and modulates the gut microbiota composition in high-fat diet fed rats. Food Funct. 2018, 1-9. ).
Growing evidence has demonstrated that gut microbiota plays an important role in the development of lipid metabolic disorders. The disturbance of the compositions of gut microbiota can disrupt the gut barrier function and hepatic cholesterol metabolism by several pathways. Previous studies have proven the impact of a high-fat diet on gut microbiota. Also, cited references (Saad, M. J.; Santos, A.; Prada, P. O. Linking Gut Microbiota and Inflammation to Obesity and Insulin Resistance [J]. Physiology, 2016, 31(4):283. Turnbaugh, P. J.; Ridaura, V. K.; Faith, J. J. et al. The Effect of Diet on the Human Gut Microbiome: A Metagenomic Analysis in Humanized Gnotobiotic Mice [J]. Science Translational Medicine, 2009, 1(6):6-6. Shang, Q.; Song, G.; Zhang, M., et al. Dietary fucoidan improves metabolic syndrome in association with increased Akkermansia, population in the gut microbiota of high-fat diet-fed mice. J. Func. Foods. 2017, 28, 138-146.).
Our study was consistent with a previous study that Spirulina platensis protease hydrolyzate (SPPH) effectively inhibited the increase of SREBP-1c expression, improved the increase of SREBP-1c expression and affected the composition of the composition of gut microbiota. So we are sure that Spirulina platensis protease hydrolyzate (SPPH) had the potential to ameliorate lipid metabolic disorders. We have added in the text. Thank you.
Reviewer 2 Report
How can authors be sure that the observed effects on lipid metabolism are due only to peptides obtained from the enzymatic treatment of S. plantensis extracts (SPEM) and not a synergic effect between peptides, γ-linolenic acid (GLA) (spirulina has an appreciable amount of GLA) and carbohydrates with potential prebiotic activity, C-phycocianin and polyphenolic compounds. These is not clear to the reviewer.
Authors fed rats with enzymatic treated extracts from Spirulina (Arthospira) platensis.
Title is misleading once authors have not used isolated peptides in their experiments.
Abstract, line 13, 17, 24 (LMD)
What is protamex?
Line 374 "The detection and identification....polymer additive-library" what is this?
Extensive editing of English language and style required
Author Response
Response to reviewer 2
1. How can authors be sure that the observed effects on lipid metabolism are due only to peptides obtained from the enzymatic treatment of S. plantensis extracts (SPEM) and not a synergic effect between peptides, γ-linolenic acid (GLA) (spirulina has an appreciable amount of GLA) and carbohydrates with potential prebiotic activity, C-phycocianin and polyphenolic compounds. These is not clear to the reviewer.
Response: Thank you very much for your positive comments. γ-linolenic acid (GLA) (spirulina has an appreciable amount of GLA) and carbohydrates with potential prebiotic activity and polyphenolic compounds were easily to dissolved organic solvent and water. Spirulina platensis powder and 95% ethanol (1:10, Wpowder : Vethanol ) were mixed at 45°C for 0.5 h, the macerate was filtered through Whatman filter paper No 3, then the supernatant was discarded in order to remove γ-linolenic acid (GLA) (spirulina has an appreciable amount of GLA) and carbohydrates with potential prebiotic activity and polyphenolic compounds, and the solid material was dried. The dried solid material was soaked in 55% ethanol at a ratio of 1:10 (w/v) at 45°C for 0.5 h, the macerate was filtered through Whatman filter paper No 3, the supernatant was then discarded in order to remove γ-linolenic acid (GLA) (spirulina has an appreciable amount of GLA) and carbohydrates with potential prebiotic activity and polyphenolic compounds, and the solid material was dried. The dried solid material was soaked in distilled water at a ratio of 1:10 (w/v) at 45°C for 0.5 h, the macerate was filtered through Whatman filter paper No 3, the supernatant was then discarded in order to remove γ-linolenic acid (GLA) (spirulina has an appreciable amount of GLA) and carbohydrates with potential prebiotic activity and polyphenolic compounds, and the solid material was dried. The dried solid material was soaked in distilled water at a ratio of 1:10 (w/v), and the pH was adjusted to 7.5 prior to the addition of compound proteinase (E:S = 1:50) and maintained at pH 7.5 and 45°C. After 1 h, the hydrolyzed solution was bathed in boiling water for 10 min to inactivate the enzyme. The supernatant was passed through a 100-µm mesh in order to remove solids in suspension. Subsequently, the supernatant was concentrated at 6,000 g and 4°C for 30 min in order to remove undigested proteins and inactivate the enzyme. So spirulina platensis proteins containing C-phycocianin were removed. Finally, the supernatant of SPPH containing the target Spirulina platensis peptides was collected and stored at -20°C prior to further analysis. The obtained dry substance was named as SPPH. And we used 95% ethanol extracted substances, 55% ethanol extracted substances, water extracted substances and SPPH to pre-test the high-fat rats, SPPH was significant affect the lipid metabolism of HFD-fed Wistar rats than three more groups the effect is not significant. So we are sure that the observed effects on lipid metabolism due to peptides obtained from the enzymatic treatment of Spirulina platensis protease hydrolyzate (SPPH).
2.Authors fed rats with enzymatic treated extracts from Spirulina (Arthospira) platensis.Title is misleading once authors have not used isolated peptides in their experiments.
Response: Thank you very much for your comments and sorry for our inaccurate expression of Spirulina platensis protease hydrolyzate (SPPH) to the reader. Now the title has been changed to “ Regulatory efficacy of the Spirulina platensis protease hydrolyzate on lipid metabolism and gut microbiota in high-fat diet-fed rats. Please see the revised manuscript.
3. Abstract, line 13, 17, 24 (LMD)
Response: Thank you very much. Lipid metabolism disorder (LMD) is blood lipid levels are out of positive range, LMD is characterized by high levels of triglyceride (TG), total cholesterol (TC) and low-density-lipoprotein cholesterol (LDL-c), coupled with low levels of high-density-lipoprotein cholesterol (HDL-c) and LMD is a risk factor for obesity, hyperlipidemia, hyperglycemia, hypertension, fatty liver, cardiopathy, clinical syndrome and other metabolic syndrome.
4.What is protamex?
Response: Thank you very much. Protamex is produced by deep liquid fermentation of Bsubtilis, and is concentrated and extracted and refined. It is a bacillus protease complex developed for hydrolyzing proteins. The main enzyme of this product is alkaline protease.
5.Line 374 "The detection and identification....polymer additive-library" what is this?
Response: Thank you very much. It is a reference to a document that has been modified in the manuscript.
6.Extensive editing of English language and style required.
Response: Thank you very much for your sincere comments and sorry very much for our poor English. We sincerely ask an expert help us with spelling and grammatical issues. Now the revised manuscript is thoroughly checked in English writing style, especially grammar and spelling. Thank you.

Reviewer 3 Report
This manuscript by Hua et al describes an investigation of the molecular effect of Spirulina platensis enzyme extractions (SPEM) in high‐fat diet (HFD)‐fed rats. The study employed a variety of biochemical assays to measure serum and hepatic levels of lipids. In addition, the effects of SPEM on the expression of genes associated with lipid homeostasis and the profile of gut microbiota were examined. This is an interesting article, which details that SPEM may have protective effects by correcting disturbances of lipid metabolism that are associated with obesity and metabolic syndrome. However there are some issues that the authors should address.
Specific Comments
Line 13. ‘Extracttions’ should be written ‘extractions’.
Lines 11-12. The sentence, ‘Spirulina platensis is a kind widely used of natural weight‐reducing agent and source of peptide’ should be re-written.
Line 14. Remove ‘In’ from ‘In our study’. Change ‘would be’ to ‘might be’.
Line 17. Remote the second full-stop after ‘liver’.
Line 23. Insert a space between the comma and ‘showed’.
Lines 33. Replace ‘result in increasing frequency’ with ‘have resulted in an increasing frequency’.
Line 37. Insert ‘the’ before ‘above-mentioned’.
Lines 59-61. Does the text need to be in italics?
Lien 80-87. Section 2.1 requires more detail. It is not clear from the text whether peptides or enzymes from SPEM are being characterized as part of the proteomic analysis. It would appear from the methods section in the manuscript that peptides were generated from Spirulina platensis proteins using Protamex.
Lines 80-87. It would be useful to include representative chromatograms and MS/MS spectra of low and high abundance peptides.
Lines 80-87. Table 1 would be better located in a supplemental section.
Line 90. The abbreviation NFD should be defined in the text as well as in the figure legend and footnotes to tables.
Lines 97-98. In the final sentence it should be clarified that the rats were fed a HFD and SPEM.
Lines 159-160. The gene expression data sets indicate that SPEM inhibits fatty acid synthesis and activates β‐oxidation in the livers of HFD‐fed rats. It would be interesting to examine the concentrations of fatty acids and acyl carnitines in the liver samples to validate these findings.
Line 170-176. Be consistent when italicising the names of bacterial families and genera.
Line 338 and 339. Figure 7 could be removed or presented in supplemental materials.
Line 359. Insert a space between ‘ethanol’ and ‘at’.
Line 365 -387. Outline whether the peptides were analyzed by liquid chromatography-tandem-mass spectrometry (LC-MS/MS) in positive or negative ion mode. In addition, how were the peptides identified - was this achieved through manual sequencing or were putative sequences searched against protein databases? This information should be provided.
Line 365-366. The final sentence of section 5.2 does not appear to have been completed.
Lines 368-369. Gas chromatography-mass spectrometry (GC-MS) is described in section 5.3 however there is no indication from the manuscript that this technique was used in the analysis of the SPEM. This point should be clarified.
Line 370. ‘rpm’ should be converted to g.
Line 371. Replace ‘and injected to UPLC system’ with ‘and injected into a UPLC system’.
Line 374-376. It is not clear as to why the leachables were identified in a vaccine. This point should be clarified.
Line 378. Replace ‘An’ with ‘A’.
Line 385. Insert a space between ‘(CID)’ and ‘using’.
Line 408. ‘rpm’ should be converted to g.
Line 413. ‘rpm’ should be converted to g
Line 418. Remove comma after the full-stop.
418-420. The final two sentences should be combined.
Author Response
Response to reviewer 3
1. Line 13. ‘Extracttions’ should be written ‘extractions’.
Response: Thank you very much. We are very sorry for our incorrect writing, according to your comment, we have revised it. Please see the correction in the text.
2. Lines 11-12. The sentence, ‘Spirulina platensis is a kind widely used of natural weight‐reducing agent and source of peptide’ should be re-written.
Response: Thank you very much. We are very sorry for our incorrect writing, according to your comment, we have revised it. Please see the correction in the text.
3. Line 14. Remove ‘In’ from ‘In our study’. Change ‘would be’ to ‘might be’.
Response: Thank you very much. We are very sorry for our incorrect writing, according to your comment, we have revised it. Please see the correction in the text.
4. Line 17. Remote the second full-stop after ‘liver’.
Response: Thank you very much. We are very sorry for our incorrect writing, according to your comment, we have revised it. Please see the correction in the text.
5. Line 23. Insert a space between the comma and ‘showed’.
Response: Thank you very much. We are very sorry for our incorrect writing, according to your comment, we have revised it. Please see the correction in the text.
6. Lines 33. Replace ‘result in increasing frequency’ with ‘have resulted in an increasing frequency’.
Response: Thank you very much. We are very sorry for our incorrect writing, according to your comment, we have revised it. Please see the correction in the text.
7. Line 37. Insert ‘the’ before ‘above-mentioned’.
Response: Thank you very much. We are very sorry for our incorrect writing, according to your comment, we have revised it. Please see the correction in the text.
8. Lines 59-61. Does the text need to be in italics?
Response: Thank you very much. Yes, text need not to be in italics.We are very sorry for our incorrect writing, according to your comment, we have revised it. Please see the correction in the text.
9. Line 80-87. Section 2.1 requires more detail. It is not clear from the text whether peptides or enzymes from SPEM are being characterized as part of the proteomic analysis. It would appear from the methods section in the manuscript that peptides were generated from Spirulina platensis proteins using Protamex.
Response: Thank you very much, We are very sorry for our unclear writing, we have revised it. Please see the text.
The mixture of Spirulina platensis powder and 95% ethanol ( 1:10, W powder : Vethanol ) at 45°C for 0.5 hour, the macerate was filtered through Whatman filter paper No 3, then discard the supernatant in order to remove γ-linolenic acid (GLA) (spirulina has an appreciable amount of GLA) and carbohydrates with potential prebiotic activity and polyphenolic compounds and dried the solid material. The dried the solid materia was soaked in 55% ethanol at a ratio of 1:10 (w/v) at 45°C for 0.5 hour, the macerate was filtered through Whatman filter paper No 3, then discard the supernatant in order to remove the γ-linolenic acid (GLA) (spirulina has an appreciable amount of GLA) and carbohydrates with potential prebiotic activity and polyphenolic compounds and dried the solid material. The dried the solid materia was distilled water at a ratio of 1:10 (w/v) at 45°C for 0.5 hour, the macerate was filtered through Whatman filter paper No 3, then discard the supernatant in order to remove γ-linolenic acid (GLA) (spirulina has an appreciable amount of GLA) and carbohydrates with potential prebiotic activity and polyphenolic compounds and dried the solid material. The dried the solid materia was distilled water at a ratio of 1:10 (w/v) , the pH was adjusted to 7.5 prior to the addition of compound proteinase (E:S = 1:50) and maintained at 7.5, 45℃. After 1 h, the hydrolyzed solution was bathed in boiling water for 10 min to inactivate the enzyme. the supernatant was passed through a 100 lm mesh, in order to remove solids in suspension, then the supernatant was concentrated at 6000g and 4°C for 30 min, in order to remove undigested proteins and inactivate the enzyme. Finnally, the supernatant was ultrafiltered, ultrafiltration was carried out in a system consisting of four tubular modules (TIA, Techniques Industrielles Appliquées, Bollene, France), with the membranes configured in series, resulting in an effective permeation area of 0.022 m2. Ceramic membranes (Pall Corporation, New York, USA) with an average pore size of 10–20 kDa were used. The process was performed at 35 ℃, with a recirculation rate of 900 L/h and transmembrane pressure of 5 bar. the supernatant containing the target Spirulina platensis peptides was collected and stored at -20 ℃ until needed for further analysis. The dry substance obtained was named as SPPH. Meanwhile, the supernatant was filtered through a hollow fiber membrane with a molecular weight cutoff of 5 kDa immediately prior to LC-MS/MS analysis.
10.Lines 80-87. It would be useful to include representative chromatograms and MS/MS spectra of low and high abundance peptides.
Response: Thank you very much. Yes, it would be useful to include representative chromatograms and MS/MS spectra of low and high abundance peptides. The objective of this study was the Spirulina platensis protease hydrolyzate (SPPH), which is a mixture of peptides. In order to more clearly characterize the mixture of peptides, we chose to present the results in a table (Table S1). According to your comments, after we separate and purify for the peptides of the Spirulina platensis protease hydrolyzate, we can do further study on the epresentative chromatograms and MS/MS spectra of low and high abundance peptides. It provides us with the useful advice in our following research, thank you very much.
11.Lines 80-87. Table 1 would be better located in a supplemental section.
Response: Thank you very much. According to your comment, we have made the table 1 as a supplement table S1. Please see the correction in the text.
12.Line 90. The abbreviation NFD should be defined in the text as well as in the figure legend and footnotes to tables.
Response: Thank you very much. according to your comment, we have revised it, NFD: normal fat diet group. Please see the correction in the text.
13.Lines 97-98. In the final sentence it should be clarified that the rats were fed a HFD and SPEM.
Response: Thank you very much. According to your comment, we have revised it. Please see the correction in the text.
14.Lines 159-160. The gene expression data sets indicate that SPEM inhibits fatty acid synthesis and activates β‐oxidation in the livers of HFD‐fed rats. It would be interesting to examine the concentrations of fatty acids and acyl carnitines in the liver samples to validate these findings.
Response: Thank you very much. It would be interesting to examine the concentrations of fatty acids and acyl carnitines in the liver samples to validate these findings, which can be done in further study. The objective of this study was the gene expression data sets, showing that Spirulina platensis protease hydrolyzate (SPPH) inhibits fatty acid synthesis and activates β‐oxidation in the livers of HFD‐fed rats, so we did not examine the concentrations of fatty acids and acyl carnitines in the liver samples. We believe it is very helpful advice to further study next step, thank you very much.
15.Line 170-176. Be consistent when italicising the names of bacterial families and genera.
Response: Thank you very much. According to your comment, we have revised it. Please see the correction in the text.
16.Line 338 and 339. Figure 7 could be removed or presented in supplemental materials.
Response: Thank you very much. According to your comment, we have made the Figure 7 as a supplement Figure S1. Please see the correction in the text.
17.Line 359. Insert a space between ‘ethanol’ and ‘at’.
Response: Thank you very much. We are very sorry for our incorrect writing, according to your comment, we have revised it. Please see the correction in the text.
18.Line 365 -387. Outline whether the peptides were analyzed by liquid chromatography-tandem-mass spectrometry (LC-MS/MS) in positive or negative ion mode. In addition, how were the peptides identified - was this achieved through manual sequencing or were putative sequences searched against protein databases? This information should be provided.
Response: Thank you very much. The peptides were analyzed by liquid chromatography-tandem-mass spectrometry (LC-MS/MS) in positive ion mode. In addition, according to your comment, we added the section of the peptides identified, which was achieved through Spirulina platensis protein databases under Uniprot.
19.Line 365-366. The final sentence of section 5.2 does not appear to have been completed.
Response: Thank you very much. We are very sorry for our incorrect writing, we have revised it. Please see the correction in the text.
20.Lines 368-369. Gas chromatography-mass spectrometry (GC-MS) is described in section 5.3 however there is no indication from the manuscript that this technique was used in the analysis of the SPEM. This point should be clarified.
Response: Thank you very much. According to your comment, we have revised it. Please see the correction in the text.
21.Line 370. ‘rpm’ should be converted to g.
Response: Thank you very much. According to your comment, we have revised it.
22.Line 371. Replace ‘and injected to UPLC system’ with ‘and injected into a UPLC system’.
Response: Thank you very much. We are very sorry for our incorrect writing, according to your comment, we have revised it. Please see the correction in the text.
23.Line 374-376. It is not clear as to why the leachables were identified in a vaccine. This point should be clarified.
Response: Thank you very much. It is a reference to a document that has been modified in the manuscript. Please see the correction in the text.
24.Line 378. Replace ‘An’ with ‘A’.
Response: Thank you very much. We are very sorry for our incorrect writing, according to your comment, we have revised it.
25.Line 385. Insert a space between ‘(CID)’ and ‘using’.
Response: Thank you very much. We are very sorry for our incorrect writing, according to your comment, we have revised it.
26.Line 408. ‘rpm’ should be converted to g.
Response: Thank you very much. We are very sorry for our incorrect writing, according to your comment, we have revised it.
27.Line 413. ‘rpm’ should be converted to g.
Response: Thank you very much. We are very sorry for our incorrect writing, according to your comment, we have revised it.
28.Line 418. Remove comma after the full-stop.
Response: Thank you very much. We are very sorry for our incorrect writing, according to your comment, we have revised it.
29.Line 418-420. The final two sentences should be combined.
Response: Thank you very much. We are very sorry for our incorrect writing, according to your comment, we have revised it.

Round 2
Reviewer 1 Report
Authors have edited the manuscript, suggested for publication
Author Response
1. Authors have edited the manuscript, suggested for publication
Response: Thank you very much for your positive comments. My article level has been further improved.
Reviewer 2 Report
Minor corrections.
line 2, Title: Spirulina platensis italics
line 8, b upper script
line 124, Spirulina platensis italic
line 308, Porphyrinaceae
line 367, Spirulina platensis italic
line 369, Spirulina platensis italic
Author Response
1. line 2, Title: Spirulina platensis italics
Response: Thank you very much. We are very sorry for our incorrect writing, according to your comment, we have revised it. Please see the correction in the text.
2. line 8, b upper script
Response: Thank you very much. We are very sorry for our incorrect writing, according to your comment, we have revised it. Please see the correction in the text.
3. line 124, Spirulina platensis italic
Response: Thank you very much. We are very sorry for our incorrect writing, according to your comment, we have revised it. Please see the correction in the text.
4. line 308, Porphyrinaceae
Response: Thank you very much. We are very sorry for our incorrect writing, according to your comment, we have revised it. Please see the correction in the text.
5. line 367, Spirulina platensis italic
Response: Thank you very much. We are very sorry for our incorrect writing, according to your comment, we have revised it. Please see the correction in the text.
6. line 369, Spirulina platensis italic
Response: Thank you very much. We are very sorry for our incorrect writing, according to your comment, we have revised it. Please see the correction in the text.
Reviewer 3 Report
The manuscript by Hua et al has incorporated a number of changes outlined in the original review. However there are a number of outstanding issues that should be addressed:
1. The manuscript would benefit from the inclusion of representative chromatograms and MS/MS spectra of low and high abundance peptides. These figures can be presented in the supplementary materials.
2. Lines 399-400. It is stated that the PEAKS 400 Studio 8 software is manufactured by Waters, MA, USA. Should the manufacturer be listed as Bioinformatic Solutions Inc, Waterloo, Canada?
3. Table S1 should provide more details. This should include whether the peptides were detected with single, double or triple charges. In addition, the theoretical mass should be listed together with the ppm mass error for each peptide.
Author Response
1.The manuscript would benefit from the inclusion of representative chromatograms and MS/MS spectra of low and high abundance peptides. These figures can be presented in the supplementary materials.
Response: Thank you very much. According to your comment, we have added the representative chromatograms and MS/MS spectra of low and high abundance peptides in the Figure S1, S2, S3, S4, S5 and S6. Please see the supplementary materials.
2. Lines 399-400. It is stated that the PEAKS 400 Studio 8 software is manufactured by Waters, MA, USA. Should the manufacturer be listed as Bioinformatic Solutions Inc, Waterloo, Canada?
Response: Thank you very much. According to your comment, we have revised it. Please see the correction in the text.
3. Table S1 should provide more details. This should include whether the peptides were detected with single, double or triple charges. In addition, the theoretical mass should be listed together with the ppm mass error for each peptide.
Response: Thank you very much. According to your comment, we have added the information in the Table S1. Please see supplementary materials.
Round 3
Reviewer 3 Report
The
manuscript by Hua et al has addressed all of the suggested modifications outlined
in the earlier reviews.